# Depollution of Polymeric Leather Waste by Applying the Most Current Methods of Chromium Extraction

**DOI:** 10.3390/polym16111546

**Published:** 2024-05-30

**Authors:** Ana-Maria Nicoleta Codreanu (Manea), Daniela Simina Stefan, Lidia Kim, Mircea Stefan

**Affiliations:** 1Department of Analytical Chemistry and Environmental Engineering, Faculty of Chemical Engineering and Biotechnology, National University of Science and Technology Politehnica Bucharest, Polizu Street, No 1-7, 011061 Bucharest, Romania; anamaria.codreanu@ecoind.com; 2Department for Evaluation, Monitoring Environmental Pollution, National Research and Development Institute for Industrial Ecology, Drumul Podu Dambovitei Street, No 57-73, 060652 Bucharest, Romania; lidia.kim@incdecoind.ro; 3Pharmacy Faculty, “Titu Maiorescu” University, 22 Dâmbovnicului Street, 040441 Bucharest, Romania

**Keywords:** leather solid waste, chromium removal, hydrolysis method, substitution method, circular economy

## Abstract

The leather industry is one of the most polluting industries in the world due to the large amounts of waste following raw hide processing but also due to the high content of chemical substances present in leather waste. The main problem with chromium-tanned leather solid waste is related to the storage, due to the ability of chromium to leach into soil or water, and also owing to the high ability of trivalent chromium to oxidize to its toxic form, hexavalent chromium. The purpose of this article is to present the most current methods of chromium extraction from solid tanned leather waste in order to obtain non-polluting leather, which can constitute secondary raw material in new industrial processes. The extraction methods identified in the present study are based on acid/basic/enzymatic hydrolysis and substitution with the help of organic chelators (organic acids and organic acid salts). In addition, this study includes a comparative analysis of the advantages and disadvantages of each identified extraction method. At the same time, this study also presents alternative chromium extraction methods based on the combination of conventional extraction methods and ultrasound-assisted extraction.

## 1. Introduction

Nowadays, industrial pollution is one of the biggest challenges all over the world. The leather industry is one of the most important industries in the world [1] and, at the same time, is known as one of the major polluting industries [2], due to the large amount of chemicals used, the large amounts of leather waste produced, and chemical compounds discharged into wastewater [3,4]. 

The leather industry is based on leather manufacturing, which is one of the oldest activities in the history of mankind [2]. This industry uses animal skins and hides as raw material [2], coming especially from the meat industry [5], to obtain a variety of products, such as belts, clothes, shoes, and bags [6]. Annually, the leather industry consumes approximately 8–9 million tons of raw skin and hide [7]. 

Because raw animal skins and hides are microbiologically unstable, it is necessary to apply a stabilization stage [8,9]. The resources used in the leather industry and the products resulting from the leather processing are described in Figure 1 [10]. 

The transformation of raw hides into a useful product called leather requires, in addition to the raw material, an energy input (process water, chemicals, process heat). This process involves pre-tanning operations, tanning operations, and post-tanning operations. Pre-tanning operations include trimming, soaking, liming, unhairing, reliming, fleshing, deliming, bating, scudding, and pickling, while tanning operations include chrome/vegetable tanning, basification, and piling. The post-tanning operations are based on sammying, splitting, shaving, rechroming, neutralization, re-tanning, fat-liquoring, dyeing, setting, and drying. Finally, the crust leather undergoes processes of conditioning, staking, spraying, coating, toggling, trimming, biffing, measuring, plating, and polishing to obtain a finished leather product [11,12,13,14].

The tanning stage is one of the most important stages of leather processing because this process transforms raw skins into non-putrescible products [15] and protects the leather against moisture, heat, and microbial degradation, offering better dyeing characteristics and softness as well [1,2,16]. In the tanning stage, chromic or vegetable compounds can be used [11]. The chromium tanning process is the most commonly used process [17,18]. In the chromium tanning process (presented in Figure 2), chromium ions are crosslinked to the carboxylic groups by the collagen present in the leather by olation and oxolation [19].

Today, over 80–90% of tanneries around the world use salts of trivalent chromium for tanning. Chromium (III) salts are among the most used tanning agents [20]. Chromium sulfate, used predominantly in the form of basic chromium sulfate (Cr(OH)(SO_4_)), is the most used chemical compound in the leather industry for tanning and processing [20]. Also, chrome alum (chrome potassium sulfate) is another chromium-based tanning agent used in the leather industry [21]. Thus, trivalent chromium (III) is one of the most important sources of leather industry pollution due to the large amount of solid waste contaminated with the chromium produced (sludge and chromium-tanned leather waste), and the large volume of contaminated wastewater generated [20]. 

As a consequence of raw hide processing, solid waste and wastewater are generated besides the final product. Leather solid waste is generated during the leather production process by applying different mechanical steps, which standardize the dimensions of products. Leather solid waste can be classified as un-tanned solid leather waste (splits, flashings, and trimmings) and organic tanned waste (shavings, splits, and trimming) [22]. During the manufacturing process of one metric ton of raw materials, the following elements are produced: 150–200 kg of finished leather, 190–350 kg of non-tanned waste, approximately 200–250 kg of tanned waste, and the rest is composed of sludge and wastewater [12,14,23,24]. 

Also, waste from leather processing operations is presented in the European Waste Catalog as *04 WASTES FROM THE LEATHER, FUR AND TEXTILE INDUSTRIES* [25].

The influence of the tannery process on the environment can be monitored by analyzing some parameters, such as total dissolved solids (TDS), chemical oxygen demand (COD), sulfates, chlorides, and heavy metal pollution (especially chrome pollution) [26]. Chromium present in water and soil samples can be detected by UV-VIS spectroscopy, inductively coupled plasma mass spectrometry (ICP-MS), and atomic absorption spectroscopy (AAS) [27].

Managing the large quantities of solid leather waste produced is a challenge for this industrial sector. The conventional methods of managing solid leather waste are disposal and incineration. Even if these methods are simple and inexpensive, they can only be applied to un-tanned solid waste, while for chromium-tanned leather waste, conventional methods are not practicable because they produce soil, water, and air contamination due to the hazardous reagents present in their composition. Chromium-tanned lather waste incineration can generate the conversion of Cr^3+^ to Cr^6+^, nitrogen oxides, ammonia emissions, and hydrogen cyanide [8,22,28]. Also, the storage of solid waste from chromium-tanned leather presents the problem of chromium leaching into the soil and underground water, which can oxidize to its toxic form Cr^6+^, affecting the health of the environment and humans. Thus, the main problem regarding the management of chromium-based solid leather waste is caused by the presence of chromium in its composition.

Chromium is an essential micro-nutrient in human and animal diets. The daily intake of chromium in the human diet is about 100 µg; it comes from cereals, fruits and vegetables, and egg yolk [29]. The recommended dose of chromium present in the diet of animals is 300 µg chromium per kg [30]. At the same time, chromium is a toxic metal for plants and microorganisms [30]. Chromium is toxic for most plants at concentrations over 100 μM·kg^−1^ dry weight [31]. This metal can exist in different oxidation states (Cr^2+^ to Cr^6+^), but in soil, it is mostly found in two oxidation states: hexavalent chromium and trivalent chromium [32]. Chromate is the most widespread form of hexavalent chromium, and the most mobile chromium form in soil [33]. Chromium presence in the two oxidation states shows different chemical and physical characteristics, in addition to opposite toxicities: trivalent chromium presents low toxicity, while hexavalent chromium presents high toxicity [34]. Trivalent chromium’s toxicity is 10 to 100 times lower than hexavalent chromium toxicity [33]. The oxidation of trivalent chromium to hexavalent chromium is pH-dependent. The possible oxidation process is described by the following equations [35]:2Cr_2_O_3_ + 8OH^−^ + 3O_2_ = 4CrO_4_^2−^ + 4H_2_O
2Cr_2_O_3_ + 2H_2_O + 3O_2_ = 2Cr_2_O_7_^2−^ + 4H^+^

In 2022, hexavalent chromium was included on the Substance Priority List issued by The Agency for toxic Substances and Disease Registry, being in the top 15 [36]. Even in small doses, hexavalent chromium affects human health, causing respiratory problems, cancer, skin ulcers, and allergic reactions [17,37], and ingested in a larger dose, it causes human and animal death [38]. A large part of solid and liquid chromium-based waste comes from the leather industry, the mining industry, the dye and pigment industry, and the textile industry [33]. Even if the chromium salts used in the leather processing process do not contain hexavalent chromium, solid leather waste contains hexavalent chromium due to the dyeing additives, pigments, and fixing agents used in finishing stages, or due to non-compliance with optimal manufacturing or storage conditions. 

Owing to the useful components present in solid tanned leather waste, alternative methods of managing thousands of tons of chromium solid leather waste are based on thermal decomposition (pyrolysis), microbial enzymatic degradation (aerobic degradation, anaerobic degradation, bioremediation), and obtaining value-added products [39]. Because chrome-tanned leather shavings contain approximately 90% collagen protein and between 3 and 6% chromium [40,41], which are valuable products, these can be reused in various fields by applying the circular economy principle (Figure 3) [42]. 

For example, due to the high content of chromium, solid waste from tanned leather can be used to produce ceramic nanopigments [43,44], or to produce re-tanning agents in the leather industry. At the same time, due to the high content of carbon and nitrogen present in the protein part, solid waste from tanned leather can be used in the production of fertilizers [45], flexible multilayer sensors [46], eco-friendly wood adhesive [47], biofuels, biochar [28,48], adsorbent materials [49,50], and construction materials [51,52].

## 2. Chromium Extraction Methods

The most common methods to significantly reduce the amounts of hexavalent chromium from waste, soils, and sludges are based on precipitation, reduction, adsorption, ion exchange, and reverse osmosis [33]. Tannery chromium (VI)-contaminated soils can be treated using microbial reduction methods in the presence of molasses, microbes, and native bacteria [53,54]. 

The recovery of chromium (III) from solid leather waste is justifiable from ecological and economical points of view, because it reduces the risk of soil, water, and air contamination. The extracted chromium can be reused, as well as other valuable components such as collagen and lipids. Chromium-tanned leather waste is the result of a covalent bond between a chromium ion and ionized carboxyl groups, belonging to aspartic and glutamic acids present in collagen [55]. Chromium–chromium bonds and chromium–collagen bonds are broken during the dechroming process of solid tanned leather waste. Dechroming can be performed by hydrolysis and by the substitution method. The most used dechroming methods are hydrolysis (acid/basic/enzymatic) and the substitution method, with the help of organic chelators [56].

Figure 4 presents the most used methods for extracting chromium from chromium-tanned leather waste. Acidic/alkaline/enzymatic extraction can be improved by applying ultrasound-assisted extraction. 

Dechroming by hydrolysis is one of the most practiced dechroming methods and can be acidic, alkaline, or enzymatic. The hydrolysis method is based on the dissociation of functional groups from the Cr–collagen matrix. Acid and alkaline hydrolysis methods have one thing in common: the protein obtained (by them) presents a low molecular weight [57]. After the hydrolysis process, a protein part (peptides or amino acids) and chromium cake can be obtained, these being valuable compounds that can be reused [58].

Depending on the hydrolysis method applied, the protein part (collagen) can be slightly denatured or strongly denatured, and it can be used in various fields, depending on the degree of hydrolysis [42]. Also, collagen extracted from animal hides or from leather waste is one of the main renewable natural resources. In animal hides and skins, the most predominant type of collagen of the 29 types is type I collagen [59]. Type I collagen is a water-insoluble protein, having a triple-helix structure and therefore presenting great industrial interest [60]. Gelatin is obtained from the partial denaturation of collagen. This important biopolymer can be extracted from solid waste leather using alkaline, acid, or a combination alkaline–acid treatment, followed by thermal hydrolysis. Figure 5 presents a scheme of gelatine production [61]. The obtained gelatin can be of two types: type A gelatin (obtained in acidic conditions) and type B gelatin (obtained in alkaline conditions) [62]. 

The hydrolysis process weakens the collagen structure, dissolves non-collagen protein, partially hydrolyzes the peptide bond, and kills bacteria. Gelatin from chromium-tanned solid leather has an excellent gel formation capacity, high biodegradability, and low cost (as shown in Table 1). The optimal extraction conditions of chromium from chrome-tanned solid waste are shown in Table 1.

## 3. Acid Extraction of Chromium

Acid methods used inorganic acids (sulfuric acid, hydrochloric acid, and azotic acid) and organic acids (oxalic acid, citric acid, tartric acid) for the dechroming of chrome-tanned solid waste [66]. Figure 6 presents the extraction of chromium from chromium-tanned leather waste with the help of an inorganic acid and an organic acid [1]. The extraction of chromium with inorganic acids, based on the acid hydrolysis method, destroys the collagen matrix while the extraction of chromium with organic chelators, based on the substitution method, keeps the collagen matrix intact.

In this method, the trivalent chromium, present in the chromium–collagen matrix, combines with the strong anionic groups of the acid, and thus, a soluble complex that can be extracted from the solution is obtained [63]. Conventionally, oxalic acid has been the best acid reagent for dechroming, followed by sulfuric acid, but the large-scale use of organic acids in the treatment of tanned leather solids is limited because of the high prices. Consequently, sulfuric acid became the first choice of many researchers, being used in numerous studies as a dechroming agent for tanned leather [63]. The main advantage of acid methods is the possibility of reusing acid solutions containing chromium in the leather industry in the tanning, pickling, and dyeing stages [66].

In acid hydrolysis, inorganic acids H^+^ can replace trivalent chromium ions and are combined with the COO^−^ of collagen. The advantages of dechroming using the acid method are stable products, low cost, and easy control and implementation [66].

The dechroming process of chromium-tanned leather samples using sulfuric acid at a low temperature was studied by varying some indicators such as sample dimensions, temperature, extraction time, sulfuric acid concentration, and sodium sulfate concentration [64]. By increasing the concentration of sulfuric acid from 35 mL/L to 200 mL/L, an improvement in the chromium extraction performance could be observed, but a high concentration of sulfuric acid led to matrix degradation. The temperature also has a similar influence on the extraction process; by changing the temperature from 296 K to 313 K, the chrome recovery yield improves, but the degradation of the protein matrix takes place. The variation in the size of the leather waste (4 mm or at sizes of 5 cm × 3 cm) did not significantly influence the chromium extraction process; the same insignificant influence was also observed in the case of varying the L-S ratio (from 5:1 to 10:1). The research team proposed two optimal chromium extraction formulas. If a high chrome recovery yield (55–60 ± 5%) is desired, but with a high degree of denaturation of the tanned leather (dissolved TOC is 3–6%), the leather waste can be subjected to a solution treatment of H_2_SO_4_ 25 mL/L, in a L-S ratio between 5:1 and 10:1, for 3–6 days at 313 K. If a moderate chromium recovery yield is desired (35 ± 5%), but with a low degree of denaturation of tanned leather (TOC < 3–6%), the leather waste can be treated with 25 mL/L H_2_SO_4_ solution, in a L-S ratio between 5:1 and 10:1, for 6 days at 296 K [64].

In addition, Wang et al. studied the complete removal of chromium from leather waste in two stages. The first stage was based on the use of the acid extraction method of chromium. The second stage of ion exchange aimed to remove the chromium found in the filtrate collected in the first stage with the help of ion exchange resins. By varying some parameters involved in the extraction process, such as the liquid–solid ratio, sulfuric acid concentration, contact time, and reaction temperature, they could observe that the factor that most strongly influences the extraction yield is the reaction temperature (50–70 °C), followed by the sulfuric acid concentration (6–10%), the liquid–solid ratio (9:1 mL/g to 11:1 mL/g), and finally, the reaction time (2.5 h to 3.5 h). By setting the liquid–solid ratio at 11:1 mL/g, the sulfuric acid concentration at 8%, the contact time at 2.5 h, and the reaction temperature at 70 °C, an efficient removal of chromium from leather waste was obtained, close to 100% [63].

However, conventional acid/base/enzymatic hydrolysis can break the peptide bonds in collagen, thus resulting in destroyed collagen, which can be difficult to remove from the hydrolysate. Chromium extraction from chromium-tanned leather waste by means of organic acids or their salts has the advantage of efficiently removing chromium without significantly degrading the collagen matrix. 

Dechroming using organic acids/salts is based on organic acid anion coordination with trivalent chromium using the substitution method (described in Figure 7). The dechroming process presents two stages. In the first stage, the breaking of the chromium–collagen bonds present in the solid tanned leather waste takes place through the substitution reaction (exchange) of a water molecule around the first chromium atom with the ionized carboxylic group belonging to the organic chelator, followed by the breaking of the first bond in the chromium–collagen to restore the coordination number of chromium. In the second stage, the ionized carboxylic groups of the organic chelator attack the second chromium atom, followed by the breaking of Cr–collagen bonds, to restore the coordination number [19]. 

The main advantages of this method are the low degree of collagen hydrolysis, high collagen recovery yields, and low amounts of dangerous materials generated [66]. At the same time, the loss of collagen from chrome-tanned solid waste using organic acid extraction is lower than alkaline methods [66].

The efficiency of chromium separation from chromium leather waste using organic acids (acetic acid and citric acid) and organic salts (potassium oxalate and potassium tartrate) was tested by Malek and colleagues [19]. The experimental results demonstrated a direct influence of some parameters involved in the process, such as the type of chelating agent used, the type of extraction medium, the concentration of the chelating agent, the concentration of the extraction medium, the temperature, and the contact time, on the chromium extraction yield (III). The tests regarding the choice of the organic chelator carried out in the neutral environment demonstrated the fact that the salts of organic acids have a higher capacity to extract chromium, compared to the organic acids used in the extraction processes. This fact is due to the presence of the alkaline environment in the case of salts, which can form stronger bonds between the carboxylic groups and chromium more easily, while the acidic environment can inhibit the reaction. Also, by testing the medium suitable for the extraction process, it was possible to conclude that, for all four organic chelating agents, the highest extraction yields were obtained in the basic medium (94.3% when potassium tartrate was used in the basic medium). By studying the effect of the potassium tartrate concentration on the extraction yield, it was observed that, by increasing the concentration of the chelating agent (0.005 M, 0025 M, 0.05 M, 0.25 M, 0.5 M, 1 M), the pH values also increased. The stabilization of the pH leads to a significant increase in the extraction yield; thus, the optimal value of the potassium tartrate concentration was 0.5 M. Also, by progressively increasing the concentration of the alkaline solution (0.01 M, 0.05 M, 0.1 M, 0.25 M, 0.5 M, 1 M), the pH tends to increase, and the Cr removal efficiency tends to follow the same trend. A concentration higher than 0.5 M leads to the degradation of the sample matrix, due to the strong manifestation of the phenomenon of hydrolysis of hydroxyl groups on the chromium–collagen complex. Through the simultaneous study of the influence of temperature (25, 30, and 40 °C) and contact time (30, 90, and 180 min), the results obtained showed that temperature and reaction time have significant effects on the efficiency of chromium extraction. A decrease in the residual chromium concentration could be observed by keeping the temperature constant at 40 °C and varying the contact time.

Also, the extraction efficiency of chromium (III) from the solid tanned leather waste with the help of organic acid salts (sodium gluconate, trisodium citrate, EDTA-2Na, sodium oxalate, sodium, and potassium tartrate) was also studied [68]. The studies about the choice of chelating agent demonstrated that all organic chelators tested show good chromium extraction capabilities, due to the presence of carboxylic groups in their molecules, which show a strong ability to complex chromium. The chromium extraction efficiency decreases with increasing chelating agent molecule size and with an increase in the carboxylic group number. The most suitable chelating agent was sodium oxalate. The existence of a directly proportional relationship between the extraction yield and the sodium oxalate concentration (1–3%) was observed. The optimal value of sodium oxalate concentration is 2%, because after this value, the phenomenon of denaturation of the collagen matrix appears. Also, the extraction yield of chromium is significantly influenced by the increase in temperature (318–338 K), and it increases with an increase in temperature, with the optimal temperature being 333 K, because above this temperature, the collagen matrix is destroyed. Further studies on the influence of the thickness of tanned leather waste samples (0.5–2.3 mm) and the stirring speed (0 to 150 rpm) demonstrated that decreasing the thickness of the waste improves the extraction yield, and increasing the speed of stirring also leads to a significant improvement of the extraction yield [68].

In addition, the influence of heteroatomic organic compounds (8-hydroxyquinoline, acetylacetone, dithizone, thiourea DMSO) and organic compounds containing a carboxyl group in the molecule (maleic acid, citric acid, oxalic acid, tartaric acid, salicylic acid, EDTA) on the dechroming process of tanned leather solid waste was studied [67]. The experimental results demonstrated that the most effective chelating agents are those containing the carboxyl group in the molecule, and the best chromium extraction yield from these wastes was obtained for oxalic acid (71%), followed by citric acid (63%) and tartaric acid (62%), the extractions being carried out in a single step. In contrast, the compounds containing sulfur atoms in the molecule (dithizone, thiourea, DMSO) showed the weakest results (0–3%). Studies on the influence of pH, temperature, and contact time on the extraction efficiency of oxalic acid and citric acid demonstrated that an increase in pH towards the neutral medium significantly improved the extraction yield for both acids, and using higher temperatures significantly decreased the extraction time and the chromium extraction yield. At temperatures exceeding 60 °C, the protein matrix in the analyzed waste was destroyed. 

Considering the advantages and disadvantages of the two methods of extraction in an acidic environment (extraction with inorganic acids and extraction with organic acids), the team coordinated by Tian studied the extraction of chromium by applying a combined acid extraction method, using a mix between an organic acid (oxalic acid) and an inorganic acid (sulfuric acid). The studies carried out followed the influence of the mass ratio between oxalic acid, sulfuric acid, and tanned leather waste but also the influence of the time interval at which the acid solution was replaced on the chrome extraction yield and on the collagen recovery yield. By varying the mass ratio of oxalic acid/sulfuric acid/leather waste in the range of 1:0.5:1–3:1.5:1, it was possible to observe that the optimal ratio is 2:1:1, as it obtained a degree of dechroming of approximately 96.5% and a collagen recovery yield of 90.6%. Also, by varying the time interval in which the acid solution was replenished, the best collagen recovery yield and the best degree of dechroming over 90% were obtained, and the solution changed 3 times in 12 h. The main advantages of the developed method included obtaining collagen with a high molecular weight, a high yield of chromium extraction, and moderate costs [66].

## 4. Alkaline Extraction of Chromium

Alkaline hydrolyses can be carried out using calcium oxide, magnesium oxide, sodium hydroxide, potassium hydroxide, and calcium hydroxide. In alkaline hydrolysis, the coordination of the chromium ion (III) to the hydroxide ion takes place, and the precipitation of trivalent chromium in Cr(OH)_3_ can be observed, while the collagen is hydrolyzed (polypeptide chains break into small peptide fragments or amino acids). Cr^3+^ formed from the soluble hydroxy complex can later be separated by extraction with another solvent (for example, quaternary ammonium compounds) [71]. The main advantages of alkaline extraction are the ease of solid–liquid mixture separation and the lack of high technology and high costs, while the disadvantages include obtaining hydrolyzed collagen with a low molecular weight and a recovery yield of protein that does not exceed 50% [66].

The study of the influence of the sodium hydroxide concentration, contact time, and temperature on the chromium (III) extraction process from the composition of tanned leather waste through the basic hydrolysis process was carried out by Wionczyk and colleagues [71]. The experimental results demonstrated that the extraction yield of chromium from the chromium–collagen matrix can be improved by increasing the concentration of sodium hydroxide from 0.2 M to 0.3 and increasing the degree of hydrolysis of the protein matrix at the same time. By increasing the temperature in the range of 313–343 K, it could be observed that a high chromium extraction yield was reached in a shorter time. Under optimal conditions of 60 °C, one-hour contact time, and 0.2 M NaOH concentration, chromium (III) was efficiently separated from the collagen–protein matrix, with the extraction yield reaching the value of 90%. However, following the application of the optimal extraction conditions, the collagen present in the solid waste from the skin was completely hydrolyzed, this aspect constituting a serious disadvantage [71].

Studies on the influence of the hydrolysis time (1–3 h) and the concentration of the extraction agent (1–3% NaOH) at a temperature of 90 °C on the results of the basic hydrolysis process of leather shaving scraps were also carried out by Pahlawan et al. [73]. The experimental results demonstrated that, by increasing the contact time to 3 h and the NaOH concentration to 3%, a hydrolysate with a protein content of 6.64% and a chromium content of 47.55 ppm could be obtained (the initial chromium content in leather shaving scraps being 23,176.14 ppm) [73]. Similar studies regarding the influence of the hydrolysis time, as well as the concentration of sodium hydroxide, on the extraction process of chromium from chrome shavings were also carried out by the research team led by Tahiri [80]. The optimal conditions were a 15 min reaction time and 0.5 M concentration of sodium hydroxide solution [80]. 

The alkaline extraction of chromium from chrome leather shavings with the help of sodium hydroxide was also studied by Barra Hinojosa and collaborators [72]. In the study, the relationship between working conditions (contact time and sodium hydroxide concentration) and the recovery yields of hydrolyzed collagen and the recovery yields of chromium, was studied. It was found that by increasing the concentration of sodium hydroxide (0.1–0.5 M), the hydrolysis reaction was improved, and the optimal concentration was 0.47 M. By varying the contact time between 30 min and 120 min, it was found that by increasing the contact time, the separation of chromium from the chromium–collagen matrix is improved, and the optimal contact time was 90 min. Thus, in optimal conditions (70 °C, 0.47 M NaOH, 90 min), 45 L of collagen hydrolysate with 87.16% protein collagen and 1.17% residual chromium were recovered [72]. 

Also, the research team of Ferreira et al. studied the chromium extraction efficiency from solid salts using sodium hydroxide as the extraction medium, and also the effects of temperature, contact time, and S-L ratio [74]. Studies of the influence of the sodium hydroxide solution concentration (0–4 M) demonstrated that the NaOH concentration is one of the most important factors in the chromium extraction process. By increasing the NaOH concentration, as in the case of the study carried out by Wionczyk, an increase in the chromium extraction yield was observed, as well as significant degradation of the collagen matrix. At the same time, studies have shown that temperature (373–473 K) and contact time (1–24 h) are important factors. Their increase leads to significant degradation of the collagen matrix (almost completely). In contrast, the S-L ratio has no significant effect on the chromium (III) extraction yield [74]. The optimal conditions were a temperature of 423 K for 1.5 h with NaOH 4 mol/L solution, and a solid-to-liquid (S/L) ratio (*w*/*w*) of 0.15 or 0.2.

In order to reduce the disadvantage of the destruction of the collagen matrix during the process of chromium extraction from tanned leather waste, the research team led by Poulopoulou implemented two additional steps to protect the protein matrix with H_2_SO_4_ and Gamma radiation, in addition to the extraction step with NaOH [75]. Also, the research carried out in the study demonstrated that the introduction of a final extraction step with hydrogen peroxide led to the improvement of the chromium extraction yield, with the final amount of chromium in the tanned leather solid waste being below 10 ppm. The study of the influence of the concentration of sulfuric acid demonstrated that an increase in the concentration of sulfuric acid (0.1–1 N) directly proportionally influences the degree of denaturation of the protein, as well as the chromium extraction yield, while the study of the influence of the Gamma radiation dose demonstrated that an increase in this parameter leads to an increase in the denaturation of the protein matrix. A concentration of NaOH higher than 1N leads to the destruction of the protein part, and a concentration below this value significantly reduces the chromium extraction yield [75]. 

Even if the extraction yields are high when alkaline extraction is used, in order to reduce the disadvantage given by the high degree of collagen hydrolysis, this method must be combined with acid hydrolysis. Thus, to improve the dechroming yield, but also to keep a low degree of collagen hydrolysis, an alternating acid–base extraction process was tried on solid waste from tanned leather [76]. Chromium extraction was carried out in four stages, as follows: basic extraction with NaOH; acid extraction with H_2_SO_4_; basic extraction with Ca(OH)_2_; and acid extraction with H_2_SO_4_. At the same time, to improve the results obtained in the study, the influence of hydrolysis assistants (ammonium hydroxide, diethanolamine, methyl-urea, guanidine hydrochloride, and urea), reaction temperature, sulfuric acid concentration, and calcium hydroxide concentration were studied. It could be observed that all the compounds with amino groups used as hydrolysis assistants were able to increase the degree of dechroming, with urea having the best capacity (65.66%). It could be observed that the largest amounts of chromium were extracted in the first two stages. By varying the concentration of Ca(OH)_2_, it was possible to observe that a high degree of chromium extraction was obtained at the optimal concentration of 40 g/L; below this value, the degree of collagen hydrolysis was lower. It has been reported that Ca^2+^ has a compressive effect on collagen fibrils at high concentrations and thereby protects the collagen fibrils from excessive damage [76].

## 5. Enzymatic Extraction of Chromium

Enzymatic extraction of chromium is based on the hydrolysis of collagen and chromium release at the same time [66]. Enzymatic hydrolysis can use different enzymes, such as crude proteolytic enzyme extracts obtained from the cultivation of new Bacillus subtilis strains, bating enzyme, and 1398 neutral protease. The advantages and disadvantages of this method are presented in Figure 8 [66].

The molecular weight of the collagen hydrolysate can be influenced by the reaction mixture and concentration of enzymes. Also, the economy of the process can be influenced by a whole series of factors, such as enzyme price, enzyme concentration, reaction speed, reaction time, and reaction yield [81]. 

Enzymatic hydrolysis does not take place individually, but together with chemical hydrolysis. Due to the limitation given by the recovery of the protein part from the obtained hydrolysate and the chromium recovery, the combination of two types of hydrolysis was tried. Enzymatic hydrolysis is performed in two steps: a chemical pre-treatment step, followed by the enzyme addition step. A pre-treatment step is mandatory in enzymatic hydrolysis because gelatin produced after the first step helps in obtaining hydrolyzed collagen in the second stage by means of enzymes [58].

For example, alkaline-enzymatic hydrolysis is composed of two stages: in the first stage, basic hydrolysis takes place (with the help of NaOH, CaO, MgO), and in the second stage, enzymatic hydrolysis is applied to the residue left from the previous stage.

Asava and collaborators tried to apply a combined alkali–enzyme hydrolysis process, to recover chrome and to improve the recovery efficiency of the protein part from tannery chrome shavings. They applied a stage of pre-treatment of the waste with MgO to denature, degrade, and increase the vulnerability of the collagen to the proteolytic attack, which followed in the next stage with the help of bating enzymes. After the first stage of basic hydrolysis with MgO 6%, the hydrolysis yield was 58.2% and the chromium content in the hydrolysate was 2.89 ppm (initial content was 3.04%). In the enzymatic hydrolysis stage, the effects of enzyme concentration (0.25–1%) and contact time (0–40 h) on protein recovery yield were studied. It could be observed that at a 0.75% enzyme concentration and 30 h contact time, the total yield of protein recovery exceeded 79.45%, and chromium was extracted in a proportion of 99.99% [77]. 

The extraction of chromium and hydrolyzed collagen from solid chromium-tanned waste combining two methods (alkaline hydrolysis and enzymatic hydrolysis) was also studied by Dettmer and colleagues [78]. In the alkaline hydrolysis stage, they studied the influence of two extraction reagents (NaOH 0.1 M and MgO) on the efficiency of the chromium and collagen protein extraction process. The experimental results demonstrated that, in the same reaction conditions (stirring of 60 rpm, 70 °C, and 15 h), a larger amount of protein was extracted by using MgO and 0.133 mg/l total chromium. In the enzymatic hydrolysis stage, the influence of two crude proteolytic enzymes, enzyme A (obtained by cultivating *Bacillus subtilis Blbc* 11), and enzyme B (obtained by cultivating *Bacillus subtilis Blbc* 17), on the chromium extraction yield and the hydrolysate extraction yield from chromium cake obtained in the first stage was studied [78]. It can be observed that the liquid extracted from the hydrolysis carried out by enzyme A had a lower chromium content (0.171 mg/ L), while enzyme B had a higher chromium content (0.359 mg/L) [78]. 

In addition, the use of two oxide mixes (CaO and MgO) in the alkaline hydrolysis step, followed by enzymatic hydrolysis with the help of 1398 neutral protease, to recover the hydrolyzed protein and chromium from chrome-tanned pig leather shavings was suggested by Qiang et al. [79]. Studies on the influence of MgO/CaO alkali dosage (0:1, 2;3, 1:1, 3:2, 1:0), alkaline hydrolysis temperature (60–100 °C), alkaline reaction time (3–7 h), enzyme dosage (0.03–0.15%), and enzyme hydrolysis temperature (42–50 °C) on the hydrolyzed protein extraction yield and on the chromium extraction yield were performed. The results showed that a hydrolyzed protein recovery yield of more than 60% was obtained when using a dosage of calcium oxide and magnesium oxide of 3%, an alkaline hydrolysis temperature of 80 °C, a time of 4 h, an enzyme dosage of 0.125%, and an enzyme hydrolysis temperature of 46 °C, while the chromium content in the final hydrolysate after desalination was less than 50 mg/Kg [79]. 

Moreover, the research team led by Rahaman carried out parallel studies regarding the influence of acid hydrolysis, the influence of acid–alkaline hydrolysis, and the influence of acid–alkaline–enzymatic hydrolysis on the chrome extraction yield from chrome shaving dust [82]. In the studies on acid hydrolysis, the influence of the type of acid extraction agent (H_2_SO_4_, HNO_3_, HCl) and their concentrations (1–6 M) was studied. Experimental studies demonstrated that HNO_3_ 6 M has the best chromium recovery capacity (70.09%). The studies on acid–alkaline hydrolysis with the help of CaO, MgO, and NaOH, and the acids H_2_SO_4_, HNO_3_, and HCl, demonstrated that all three alkaline media had nearly the same ability to recover chromium, but the mixture of MgO and HNO_3_ presented the best extraction capacity of chromium (43.71%). Following the studies on acid–alkaline–enzymatic hydrolysis carried out with the help of the three acids, the three bases listed previously, and proteolytic enzymes, it was concluded that this technique significantly improves the chromium extraction yield. The best chromium extraction yield was 55.51% when MgO, NHO_3_, and proteolytic enzymes were used. 

## 6. Ultrasound-Assisted Extraction of Chromium

Chromium extraction processes that are based on enzymatic treatment or alkaline–acid treatments are strongly affected by the formation of metabolic inhibitory products and by the high costs of purchasing pure enzymes in the case of enzymatic hydrolysis, and by the use of concentrated reagents, numerous steps, and a relatively long reaction time (72 h) in the case of acid–alkaline treatments. In addition, both processes lead to significant degradation of the protein matrix in the hide waste, reducing the quality of the raw material. Considering the disadvantages listed above, ultrasound-assisted extraction of chromium can be a sustainable alternative to existing methods for the extraction of chromium from leather waste. Ultrasound-assisted extraction can be used as an additional method to the classic extraction methods (basic, enzymatic, acid hydrolysis, and substitution with organic chelators) in order to improve the extraction yield. Studies have shown that the additional application of ultrasound in the chromium extraction process from solid tanned leather waste significantly improves the chrome extraction yield from this waste, due to the effects caused by acoustic cavitation [69].

The ultrasound propagation process involves cavitation bubbles, vibrations, crushing, mixing, and other comprehensive effects in the liquid medium [83]. When ultrasound is applied in extraction processes in a liquid medium containing solids, at the liquid−solid interface, bubbles can be formed with further collapse (cavitation), increasing the surface area of the substrate, as well as the transport rates of extraction solution [84]. The main advantages of ultrasound-assisted extraction include a short extraction time but also the use of mild extraction conditions (temperature, concentration, and amounts of reagents) [85]. 

The results of the study carried out by Bizzi and colleagues highlighted the fact that, by the additional application of an ultrasonic stage in the chromium extraction process in an acidic medium (HCl, NHO_3_, H_2_SO_4_, CH_2_O_2_, C_2_H_2_O_4_), the chromium extraction yield was significantly improved [65]. Studies revealed that ultrasound-assisted extraction leads to a chromium extraction yield of 92%, while the application of the agitation step leads to a chromium extraction yield of 65%. Also, the studies related to the influence of some operating parameters, such as the type of extraction solution, the concentration of the extraction agent, the temperature, the time, the amount of the sample, and the amplitude, demonstrated the existence of a direct relationship between the extraction yield of chromium (III) and these parameters. By testing five inorganic acids used as chromium extraction media (HCl, NHO_3_, H_2_SO_4_, CH_2_O_2_, C_2_H_2_O_4_), it was revealed that nitric acid had the highest chromium extraction capacity, owing to its oxidation power and demineralization potential, followed by sulfuric acid. By increasing the concentration of nitric acid (0.1–4 mol/L), a significant increase in chromium extraction yield could be observed because the high proton concentrations of HNO_3_ destabilized the strong Cr-Cr bonds between neighboring chromium atoms in the chromium-collagen matrix, but after a certain concentration, a destabilization of the protein structure occurred. The study of the parameters directly involved in the ultrasound stage demonstrated that there is an increase in the chromium extraction yield at low frequencies (25–37 KHz) using the ultrasound bath, but also an increase in extraction with an increase in the applied amplitude (10–90%). Studies on the influence of temperature (10–90 °C) have shown that there is an improvement in the extraction yield with increasing temperature, but above 30 °C, degradation of the protein matrix occurs, as it is well known that collagen denaturation occurs at 37 °C [65]. 

The influence of the additional application of the ultrasound process to the chemical extraction process of chromium with EDTA was studied [69]. The results demonstrated that by applying ultrasound-assisted extraction, the total extraction time (including extraction and washing steps) of chromium found in the waste composition was reduced to 45 min. In addition, it was possible to conclude that the extraction yield is significantly influenced by the variation of other parameters involved in the extraction process of chromium from tanned leather waste (extraction temperature, L-S ratio, extraction time). The study of the influence of the S:L ratio (Cr^3+^: EDTA) showed that the extraction yield improves significantly when the ratio increases (in the range of 1:0–1:6), but the optimal value is 1:3, because in the case of a ratio of 1:6, a similar yield is obtained. The chromium extraction yield follows a similar trend: when the extraction temperature is varied (60 °C to 90 °C), but at temperatures higher than 80 °C, a degradation of the skin waste is observed, while the variation of the ultrasonication time (30–120 min) does not significantly influence the yield [69].

At the same time, a similar study on the influence of ultrasound in the presence of an organic acid salt (EDTA) on the extraction yield of Cr (III) was carried out by Pedrotti and collaborators [70]. By studying the influence of the ultrasonic power, the applied temperature, and the contact time, it was possible to observe a direct influence between increasing the values of these parameters and the amount of chromium extracted from the analyzed leather waste. Increasing the temperature (30–80 °C) led to an increase in the extraction yield, but at temperatures higher than 70 °C, a degradation of the solid skin waste was observed, turning it into a gelatinous fraction. Also, a decrease in contact time and non-breakage temperature to achieve high extraction yields was observed when the ultrasonic power increased. By varying the washing time (3–10 min), an insignificant influence on the extraction yield was observed [70].

## 7. Conclusions

The purpose of this review was to present the most current methods used for dechroming solid chrome-tanned leather, in order to reuse it as an alternative raw material in new industrial sectors by applying the principles of circular economy. 

In this study, the latest acid, basic, and enzymatic extraction techniques for trivalent chromium were identified based on the hydrolysis method or on the substitution method, and at the same time, the advantages and disadvantages of each identified method were also discussed. It could be observed that by using the hydrolysis method to extract chromium from tanned leather waste, chromium recovery yields between 30 and 100% can be obtained, but at the same time, the collagen matrix can be partially (3–10%) or totally (~100%) destroyed. On the other hand, using the substitution method with the help of organic chelators, high chromium extraction yields (over 71%) were obtained without significantly degrading the collagen matrix, but the purchase costs of the organic reagents used were relatively higher. In addition, it could be observed that by applying the ultrasonic process in chromium extraction experiments, the extraction time, concentration, and amount of reagents were significantly reduced, while the chromium extraction yields exceeded 71.7%.

Based on the literature studies carried out, it is possible to conclude that a combination of the hydrolysis and substitution methods with the ultrasound process leads to significantly improved chromium removal yields from leather solid waste and a low collagen matrix degradation yield. 

## Figures and Tables

**Figure 1 polymers-16-01546-f001:**
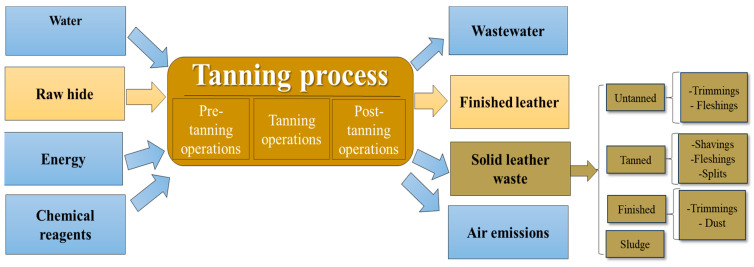
Tanning process: resources used and waste generated.

**Figure 2 polymers-16-01546-f002:**
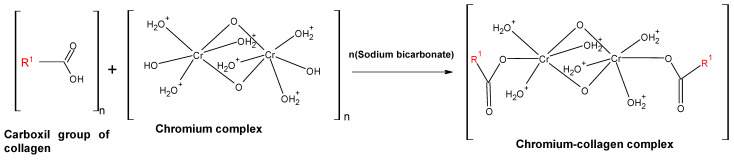
Description of the chrome tanning process of leather [19].

**Figure 3 polymers-16-01546-f003:**
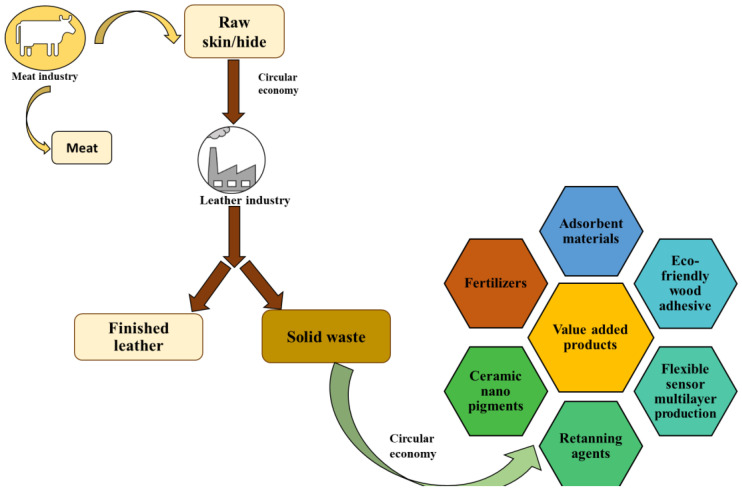
Valorization of solid leather waste by applying circular economy principles.

**Figure 4 polymers-16-01546-f004:**
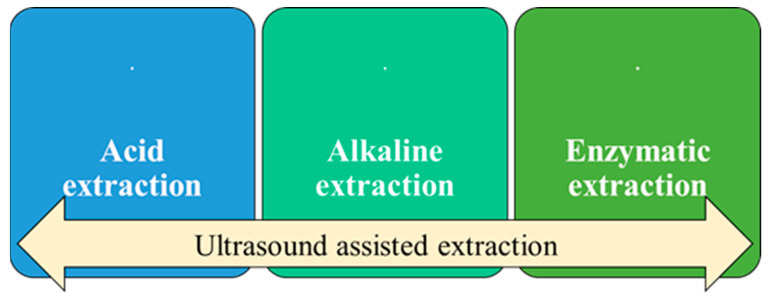
The most used chromium extraction methods from chromium-tanned leather waste.

**Figure 5 polymers-16-01546-f005:**
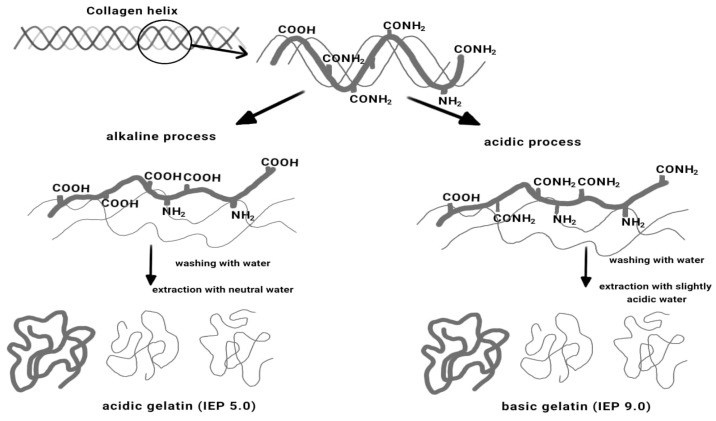
Scheme of acidic gelatin and basic gelatin production [61].

**Figure 6 polymers-16-01546-f006:**
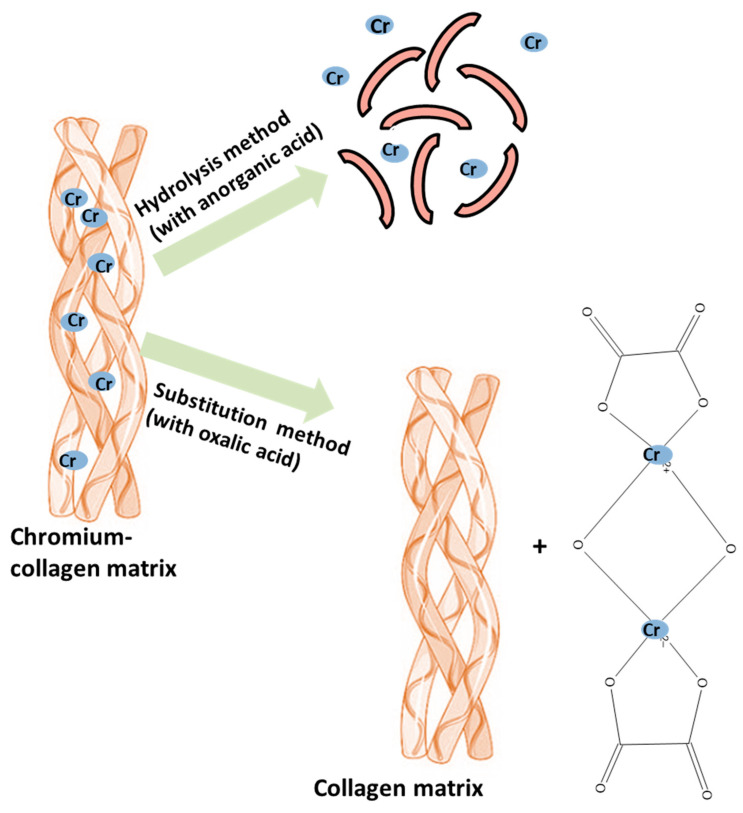
Extraction of chromium by hydrolysis method and substitution method.

**Figure 7 polymers-16-01546-f007:**
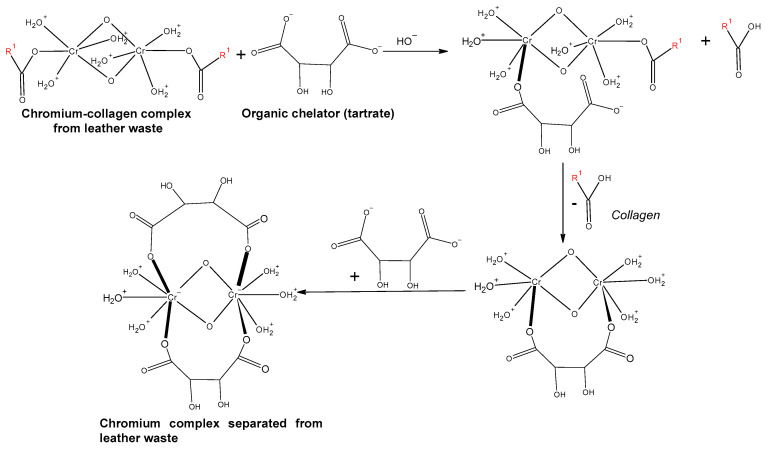
Description of leather waste dechroming process using the substitution method [19].

**Figure 8 polymers-16-01546-f008:**
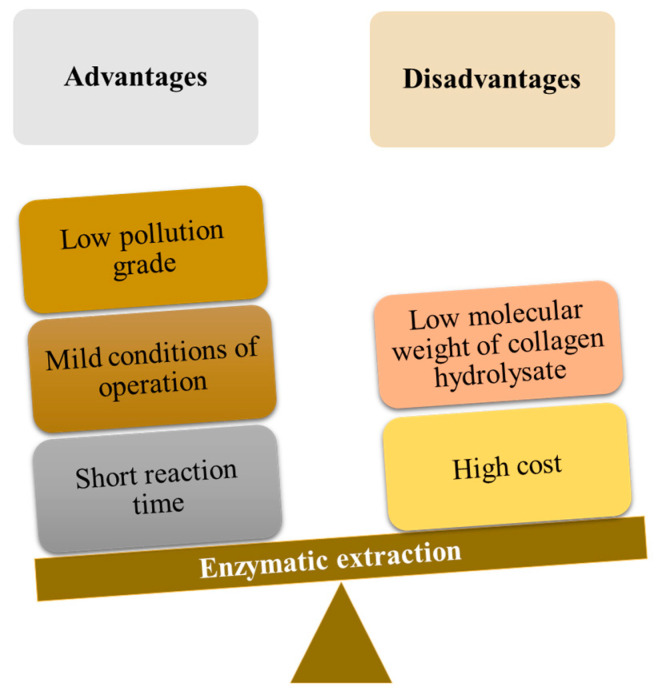
The main advantages and disadvantages of the enzyme extraction of chromium from chromium-tanned solid wastes.

**Table 1 polymers-16-01546-t001:** Optimal extraction conditions of chromium from chrome-tanned solid waste.

Chromium Removal Method	Chromium Removal Method	Chromium Extraction Yield	The Degree of Collagen Hydrolysis	Reference
Acid extraction	- Concentration of extraction solution = 8% H_2_SO_4_ - H_2_SO_4_:sample ratio = 11:1 - Time = 2.5 h - Temperature = 343 K	>95%	-	[63]
Acid extraction	- Concentration of extraction solution = 25 mL/L H_2_SO_4_	30–60% ± 5%	3–6 ± 1%	[64]
	- Time = 3 or 6 days			
Acid extraction	- Sample amount = 150 mg	92%	-	[65]
	- Concentration of extraction solution = 3 mol/L HNO_3_			
	- Temperature = 30 °C			
	- Time = 30 min - Amplitude = 90% US bath = 37 kHz			
Acid extraction	- H_2_C_2_O_4_/H_2_SO_4_/sample ratio = 2:1:1 - Time = 12 h - Processing times = 1 h - Stirring speed = 250 r/min - Temperature = 40 °C	95.6%	90.6%	[66]
Acid extraction	- Extraction agent: oxalic acid - Time = 36 h - Room temperature - pH = 5.5 - Cr–oxalic acid ratio = 1:3	71%	-	[67]
Acid extraction	- Concentration of potassium tartrate = 0.5 M - NaOH solution concentration = 0.25 M - Room temperature - Time = 72 h	95%	-	[19]
Acid extraction	- Concentration of sodium oxalate = 2% - Sodium oxalate/sample ratio = 200 mL/g - Thickness of sample = 0.5 mm - Temperature = 333 K - Time = 5 h - Stirring speed = 150 rpm	98%	>95%	[68]
Acid extraction	- Sample amount = 3 g - Cr^3+^/EDTA ratio = 1:3 - Temperature = 80 °C - Time = 30 de minutes - Ultrasonic bath = 25 kHz Amplitude = 100% - 5 washing cycles with water (V = 50 mL), at a temperature of 50 °C, for 3 min each	98%	-	[69]
Acid extraction	- EDTA/Cr ^3+^ ratio = 3:1 - US power = 150 W - Frequency = 20 KHz - Residence time = 60 min - Temperature = 70 °C	71.7%	-	[70]
Alkaline extraction	- Concentration of extraction solution = 0.2 M NaOH - NaOH/sample ratio = 80 cm^3^/g - Time = 1 h - Temperature = 60 °C	90%	~100%	[71]
Alkaline extraction	- Concentration of extraction solution = 0.47 M NaOH - Time = 90 min - Temperature = 70 °C	750.8 g	87.165%	[72]
Alkaline extraction	- Concentration of extraction solution = 3% NaOH - Time = 180 min - Temperature = 90 °C - NaOH/sample ratio = 5:1	~100%	-	[73]
Alkaline extraction	- Concentration of extraction solution = 4 M NaOH - NaOH/sample ratio = 0.15 - Time = 90 min - Temperature = 423 K	85%	98%	[74]
Alkaline extraction	- H_2_SO_4_ concentration = 0.1 N H_2_SO_4_ - Dose of Gamma radiation ^60^Co = 60 Krad - Concentration of extraction solution = 1 N NaOH	~100%	25–40%	[75]
Alkaline extraction	- Concentration of extraction solution 1 = 2 g/L NaOH, stirring for 30 min at 30 °C, urea concentration = 40 g/L - Concentration of extraction solution 2 = 50 g/L H_2_SO_4_, stirring for 1 h at 30 °C - Concentration of extraction solution 3 = 40 g/L CaOH, stirring for 2 ore at 30 °C - Concentration of extraction solution 4 = 50 g/L H_2_SO_4_, stirring for 1 h at 30 °C	97%	10%	[76]
Enzymatic extraction	- Extraction solution concentration = 6% MgO - Bating enzyme concentration = 0.75% - Time = 30 h - Temperature = 33–37 °C - pH = 8.3–8.5	99.99%	-	[77]
Enzymatic extraction	- Extraction solution MgO - Stirring speed = 60 rpm - Temperature = 70 °C - Time = 6 h - *Bacillus subtilis enzyme A proteolytic* activity = 130.5 U/mL - pH = 9 - Time = 15 h - Temperature = 45 °C - Stirring speed = 60 rpm	~100%	-	[78]
Enzymatic extraction	- Extraction solution concentration = 3% MgO - Extraction solution concentration = 3% CaO - Temperature = 80 °C - Time = 4 h - 1398 neutral protease concentration = 0.125% - Temperature = 46 °C	~100%	>60%	[79]

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
