# Peer review of "Depollution of Polymeric Leather Waste by Applying the Most Current Methods of Chromium Extraction"

_polymers, 2024, doi:10.3390/polym16111546_

Round 1

Reviewer 1 Report

Comments and Suggestions for Authors

Authors need to address following issues before moving further

1. Title  needs to be revised

2. Authors need to explain what is the novelty of this review that make it different from already written reviews

3. What is the profile of first author (please send CV) that make him/her knowledgeable  enough to write review on this topic

4. Components from Instructions to authors should be removed from manuscript. for example in start of section 2. Chromium extraction methods and end of section 6. Ultrasound assisted extraction of chromium

5. There is a need to improve the English language: there are many instances of incorrect English language so comprehensive check of manuscript is required

Examples:

 chemicals present in their composition.

The main problem of chromium-tanned leather solid waste is related to their storage,

Because raw animal skins and hides are microbiologically unstable, it is necessary to implement a stabilization stage

In 2022, hexavalent chromium is included 

chromium, d because of the dyeing additive

, it was possible to observe a directly proportional influence  between the increase in the values of these parameters and the amount of chromium

6. Redraw figure 5 to depict what it is relating

Comments on the Quality of English Language

After the correction in English language, I can asses the manuscript in detail

5. There is a need to improve the English language: there are many instances of incorrect English language so comprehensive check of manuscript is required

Examples:

 chemicals present in their composition.

The main problem of chromium-tanned leather solid waste is related to their storage,

Because raw animal skins and hides are microbiologically unstable, it is necessary to implement a stabilization stage

In 2022, hexavalent chromium is included 

chromium, because of the dyeing additive

, it was possible to observe a directly proportional influence  between the increase in the values of these parameters and the amount of chromium

Reviewer 2 Report

Comments and Suggestions for Authors

The most used methods of chrome extraction from leather polymer wastes is a review, that has an classical design for this type of article and it is included introduction, several methods of chrome extraction in the main parts and conclusion section. This review discussed main method of Chromium extraction and collagen production. While the article provides some interesting information in this field, some details could be improved.

1. Line 33. Please, provide refs in following format [1] (not (1)).

2. Line 50-52, your statement “In the tanning stage, chromic or vegetable compounds can be used”. I think this statement could be improved by adding statistic date how many manufactures are used chrome (III) as tanning agent? I think analysis how many manufactures carry out tanning stage with chrome  will support actuality of your research.

2. line 54, you point on only one form of chrome tanning agent, that is Chromium (III) salt, it only one compound that is used in tanning process or there are other salt that is used? Please point other Chromium (III) salt.

3. Figure 1. I think this scheme need more explanation, what kind energy sources are used? Also tanning process are not discussed from this scheme. What include tanning and pre-tanning process.

4. lines 74-86, This is copywriting from European Waste Catalog. I think this lines could be removed and provide only index 04 in the text.

5. lines, 74-75. Why do your use abbreviate CED for European Waste Catalog, not EWC? Is it single used of this abbreviate? If yes, please remove abbreviate.

6. lines 87-89, This indicators are very common, the main discussed pollutant in your research is a chrome. I think It’ll be better to provide recent methods for chrome detection.

7. Paragraph with lines 90 and 103. If you provided the Cr as pollutant scheme, than will show changes of the Cr form after tanning process and soil, water migration it’ll be significant improve the article.

8. line 114, please check the charge of the following ion it should be 2- (CrO42- is correct)

9. line 115, please check this eq. the right part it should be 2Cr2O72-. Please, correct index before ion and charge of the ion.

10. I think, reaction with Cr during tanning process should be added into the manuscript.

9. lines 145-159, must be removed.

10. lines 168-169, The purpose of the article should be given in introduction parts.

11. figure 3 are not informative, but if you support with statistical date, how many factories used acid extraction or alkaline extraction or enzymatic extraction? Other methods should be added with the per cent of using. According this fig this methods are used with the same frequency. What about Ultrasound assisted extraction? Why don’t you add it to the overall discussion on this fig?

12. lines 178-183. What form of chromium do you mean in this paragraph? Chromium (VI) are not hydrolyze, Chromium (III) is not toxic (lines 110-111). Chromium (III) could be hydrolyzed by alkaline compound and this form could be collected as Cr(OH)3 sediment. How does acid hydrolyze occur? Cr3+ will be sediment with anion of acid, but this process is sedimentation, not hydrolyze or Chromium (III) treatment occurs in other way?

13. lines 186-200, it should be provide scheme of Gelatin production with chemical reaction or scheme for better visualization of the process and understanding Cr transformation in this process.

14. table 1. Please give full name of L-S ratio abbreviator.

15. line 209, which kind of  chromium form combines with strong acid groups? Is the strong acid groups are anions of the acid? Before acid treatment what kind of form of Chromium was present? Is it was soluble?

16. figure 4 is not informative. It’ll be better to give scheme of Cr (I suppose Cr(III)) treatment with H2SO4 and with organic acid. How change mechanism of Cr removing after adding organic acid?

17. Page 8-10 is the description of the main reports of Acid extraction. What kind of method are most effective for now days? What kind of researches should be done in this field to improve Cr treatment by this method? What kind of disadvantages are in this methods?

18. The same comment for Alkaline extraction section. Could your summaries all protocols and give the best way on your opinion how minimization the effect on collagen?

19. As for Enzymatic extraction section. It should be given an information of what kind of enzymes are used. Scheme of the discussed process should be added. And at the end of section it should be given an opinion is it reasonable to use enzymes how to decrease of disadvantage of the Enzymatic extraction processes. And the main question how Cr removing from this process like in Alkaline extraction? Should Enzymatic extraction should be separate into a methods or it’s a part of Alkaline extraction?

20. Ultrasound assisted extraction of chromium section should be explain why Ultrasound doesn’t effect on collagen, and why this step increase the chrome extraction yield? Is this stage are used with acid extraction or can be used with alkaline extraction?

21. lines 574-576 must be removed.

The manuscript are provide interesting information, but this review is lack of explanation of the chemical process, some information are given the text by if it’ll be visible by chemical scheme or reaction it’ll be significant improved the article. This correction should be done before the review will be published.

Round 2

Reviewer 1 Report

Comments and Suggestions for Authors

Following corrections are needed

Ultrasound assisted extraction of chromium should be included in Table 1

Line 25: composition of solid waste from tanned leather

replace as

solid tanned leather waste

Line 38 replace “from” the world with “of” the world

“in” the same time  with “at” the same time

Line 93 delete “that have the role of uniformity”

Line 134-135: Write with proper clarity “Chromium is an essential micro-nutrient in human and animal diet, and/ being in 134 the same time, a toxic metal for plants and microorganisms” and also explain what levels of chromium are permissible for humans and animal dies?

Line 286-289: Rewrite sentence “In addition, Wang and co. studied the complete removal of chromium from chro-mium leather waste in two steps, using the acid extraction chromium method and ion exchange step, for removing the chromium found in the composition of the filtrate with the help of ion exchange resins.”

Line 616-617: Correct  “Also, the influence of ultrasound in presence of on chromium extraction used an 616 organic acid salt (EDTA) was studied”

Line 619: Explain that 45 minutes are reduced as compared to sample without radiation or it was the total time of extraction needed.

Comments on the Quality of English Language

English is fine now

Some minor corrections are needed as given in comments

Reviewer 2 Report

Comments and Suggestions for Authors

The authors carried out significant improvement of the manuscript, but the quality of the new added figures should be increased, especially for fig.2 and fig. 7. For using fig. 6, please, add information about right (Copyright © 2020, ...) and  use correct link (https://doi.org/10.1016/j.wasman.2019.12.039).
